# EllieSQL:
# Cost-Efficient Text-to-SQL with Complexity-Aware Routing

**Yizhang Zhu**[1*]**, Runzhi Jiang**[1*]**, Boyan Li**[1]**, Nan Tang**[1,2]**, Yuyu Luo**[1,2†]
[1]The Hong Kong University of Science and Technology (Guangzhou), Guangzhou, China
[2]The Hong Kong University of Science and Technology, Hong Kong SAR, China

## Abstract

Text-to-SQL automatically translates natural language queries to SQL, allowing non-technical users to retrieve data from databases without specialized SQL knowledge. Despite the success of advanced LLM-based Text-to-SQL approaches on leaderboards, their unsustainable computational costs—often overlooked—stand as the "elephant in the room" in current leaderboard-driven research, limiting their economic practicability for real-world deployment and widespread adoption. To tackle this, we exploratively propose EllieSQL, a complexity-aware routing framework that assigns queries to suitable SQL generation pipelines based on estimated complexity. We investigate multiple routers to direct simple queries to efficient approaches while reserving computationally intensive methods for complex cases. Drawing from economics, we introduce the Token Elasticity of Performance (TEP) metric, capturing cost-efficiency by quantifying the responsiveness of performance gains relative to token investment in SQL generation. Experiments show that compared to always using the most advanced methods in our study, EllieSQL with the Qwen2.5-0.5B-DPO router reduces token use by over 40% without compromising performance on Bird development set, achieving more than a $2\times$ boost in TEP over non-routing approaches. This not only advances the pursuit of cost-efficient Text-to-SQL but also invites the community to weigh resource efficiency alongside performance, contributing to progress in sustainable Text-to-SQL. Our source code and model are available at https://elliesql.github.io/.

## 1 Introduction

Text-to-SQL converts natural language into SQL queries, significantly lowering the barriers to database interaction and advancing the democratization of data science (Qin et al., 2020; Luo et al., 2018). With the advent of large language models (LLMs) and their powerful semantic understanding capabilities, LLM-based Text-to-SQL approaches have demonstrated remarkable performance on prominent benchmarks (Liu et al., 2024; Li et al., 2024b; Yu et al., 2018). These methods often require sophisticated and computation-intensive SQL generation processes, including fine-grained task divisions (Talaei et al., 2024), online synthesized examples and divide-and-conquer approaches (Pourreza et al., 2025), and Monte Carlo Tree Search-enhanced reasoning (Li et al., 2025).

Despite the success of these advanced LLM-based Text-to-SQL solutions, there is a crucial challenge: excessive token consumption. While these approaches achieve high ranking on leaderboards, the performance improvements are marginal relative to their exponential computational demands. When handling massive volumes of queries in real-world applications, the computational expenses associated with sophisticated pipelines can render such systems economically unsustainable. This issue represents a fundamental yet frequently overlooked limitation, posing a major obstacle to deployment outside the laboratory. We argue this is an **"elephant in the room"** for the current leaderboard-driven Text-to-SQL research.

---

[*]Equal contribution.
[†]Yuyu Luo is the corresponding author. E-mail: yuyuluo@hkust-gz.edu.cn

A notable observation is that existing approaches apply a uniform processing pipeline to all queries, despite the fact that not all queries necessitate such sophisticated workflows to generate correct SQL. Even basic baseline models, without any reasoning enhancements, are capable of solving a fair number of cases. For many simple queries, indiscriminately applying a complicated process not only results in a large amount of unnecessary token overhead but may also lead to overthinking, increasing the risk of errors in straightforward cases (Cuadron et al., 2025).

To address it, we propose EllieSQL, a complexity-aware routing framework for SQL generation. EllieSQL employs a three-tiered SQL generation module (Basic, Intermediate, and Advanced) that dynamically routes queries to the appropriate tier based on their estimated complexity. By efficiently handling simpler cases with lightweight SQL generation methods and reserving computationally intensive pipelines for genuinely complex ones, EllieSQL significantly enhances cost-efficiency while maintaining high performance.

To quantitatively assess the cost-efficiency of SQL translation, we introduce Token Elasticity of Performance (TEP), a metric grounded in economic principles (Marshall, 2013). TEP treats tokens as a form of computational investment and quantifies the responsiveness of performance improvements relative to increased token consumption. This provides a standardized measure to evaluate the cost-efficiency trade-offs inherent in different SQL generation approaches, which is particularly crucial for real-world deployment scenarios where both performance and computational efficiency are essential considerations.

We aim to invite the whole community for further research into balancing performance with resource expenditure, hoping to contribute to more practical and sustainable Text-to-SQL. This paper serves as an exploration, a stepping stone rather than a definitive answer, intended to spark further research and discussion. In this paper, we try to *confront the elephant* with contributions summarized as follows:

- **Elephant in the Room.** We highlight a critical yet overlooked cost-efficiency limitation in current Text-to-SQL methods, hindering their practical deployment beyond laboratories.
- **TEP Metric.** We introduce Token Elasticity of Performance (TEP), an economic-inspired metric for evaluating the responsiveness of performance gains relative to token investments in SQL generation.
- **EllieSQL.** We propose EllieSQL framework with various router implementations to direct queries to appropriate tiered SQL generation pipelines based on estimated complexity.
- **Extensive Experiments.** Experiments exhibit the potential and effectiveness of EllieSQL with various routers. Notably, with the Qwen2.5-0.5B-DPO router, we reduce token use by over 40% without sacrificing performance compared with consistently deploying the most advanced SQL generation pipeline, achieving more than a $2\times$ boost in TEP.

## 2 Related Work

**Text-to-SQL.** Text-to-SQL converts natural language queries into SQL statements. Since large language models (LLMs) have demonstrated remarkable capabilities and versatility in data analysis (Zhu et al., 2024; Wu et al., 2025), researchers have increasingly explored their potential in Text-to-SQL tasks (Chen et al., 2024a; Liu et al., 2025). Earlier work, such as DAIL-SQL (Gao et al., 2024), has systematically benchmarked LLM-based Text-to-SQL. Li et al. (2024a) propose NL2SQL360, a testbed for comprehensively evaluating Text-to-SQL methods from multiple angles. DIN-SQL (Pourreza & Rafiei, 2023) performs internal classification, heuristically categorizing queries into manually-defined categories to select category-specific few-shot examples and handcrafted prompts to improve accuracy.

More recent efforts emphasize fine-grained task decomposition and advanced reasoning. CHESS (Talaei et al., 2024) meticulously designs a highly refined schema pruning technique. CHASE-SQL (Pourreza et al., 2025) enhances the SQL generation with three Chain-of-Thought strategies: divide-and-conquer, online synthesis, and query planning. Monte Carlo Tree Search has also been utilized to strengthen reasoning in Text-to-SQL systems like Alpha-SQL (Li et al., 2025), which iteratively infers SQL construction actions based on partial SQL query states. YORO (Kobayashi et al., 2025) internalizes database schema into the parametric knowledge of LLMs via fine-tuning to reduce input tokens in Text-to-SQL.

However, these approaches uniformly apply computationally intensive techniques to all queries, regardless of their difficulty. The inefficiency of overprocessing simple queries that lightweight methods could handle uncovers an opportunity for optimization via complexity-aware routing.

**LLMs Routing.** Routing trains a model to select the most suitable LLM for each query without running inference on all candidates (Zhang et al., 2024; Xiang et al., 2025). RouteLLM (Ong et al., 2025) fine-tunes routers to make binary routing decisions via a classification token, while ZOOTER (Lu et al., 2024) distills rewards from training queries to guide routing. RouterDC (Chen et al., 2024b) employs dual contrastive learning, Hybrid LLM (Ding et al., 2024) assigns queries based on predicted difficulty and desired quality level. Malekpour et al. (2024) conduct preliminary experiments on routing for a proper LLM out of three for SQL generation yet sacrificing performance. LLM cascading offers a sequential alternative to assign tasks based on confidence scores post-generation, yet increasing costs for complex queries (Varshney & Baral, 2022). While existing LLM-level routing methods primarily focus on selecting models for each query, Text-to-SQL research reveals that optimized reasoning pipelines play a fundamental role in achieving performance improvements in SQL generation. Thus, the strategic selection of suitable SQL generation pipelines is fundamental, but complexity-aware routing for them remains unexplored.

## 3 Overview of EllieSQL Framework

In this section, we will first present the overview of EllieSQL framework and its three key phases. Following this, we discuss the metrics involved to assess Text-to-SQL and routers' performance, as well as the newly introduced Token-Elasticity of Performance (TEP) for evaluation on cost-efficiency in SQL generation.

### 3.1 Overview

Given a database $\mathcal{D}$ and a user's natural language query $\mathcal{Q}$, the goal of Text-to-SQL is to produce a SQL statement $y$ based on $\mathcal{D}$ and $\mathcal{Q}$. Formally, $y = f(\mathcal{D}, \mathcal{Q})$. To achieve this goal efficiently across varying query complexities without compromising performance, we propose EllieSQL, a complexity-aware routing framework for cost-efficient Text-to-SQL.

As shown in Figure 1, our EllieSQL operates in three key phases: schema linking, routing, and tiered SQL generation pipelines. At the heart of EllieSQL lies the router, serving as the *decision-making core*, which dynamically directs queries to the appropriate tier of SQL generation pipelines based on estimated query complexity.

Our motivation stems from the following observation: in practice, Text-to-SQL tasks exhibit significant heterogeneity in complexity, where not all queries require the most sophisticated and resource-intensive methods for effective resolution. Existing leading approaches, however, indiscriminately apply complex reasoning and computationally expensive techniques to all queries, resulting in inefficiency as simple queries could be adequately addressed by lightweight methods. Moreover, Text-to-SQL research reveals that enhancing reasoning within SQL generation is fundamental for improving performance, particularly for complex queries, which has driven the development of diverse, meticulously designed SQL generation pipelines. Such diversity underscores a clear opportunity for complexity-aware routing at the pipeline level, which promises to optimize computational efficiency while providing enterprises with the flexibility to integrate tailored pipelines for their specific requirements. Next, we elaborate on the overview and formulation of each phase within EllieSQL.

### 3.2 Phase I: Schema Linking

**Overview.** Given a database $\mathcal{D}$ and natural language query $\mathcal{Q}$, the database schema $\mathcal{S}_\mathcal{D}$ often contains numerous elements irrelevant to $\mathcal{Q}$. Schema linking is the process of identifying the tables and columns necessary for SQL generation aligned with the user's intended query (Lei et al., 2020; Liu et al., 2024). By eliminating irrelevant schema elements, schema linking reduces input complexity and prevents LLMs from being overwhelmed with redundant information that could lead to performance degradation, making schema linking a critical component in Text-to-SQL (Cao et al., 2024; Talaei et al., 2024; Li et al., 2025).

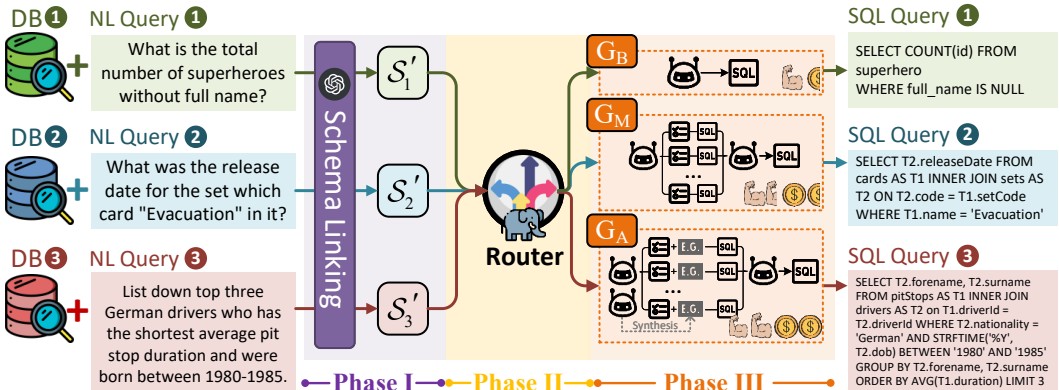

Figure 1: Framework of EllieSQL, consisting of three phases: 1) schema linking; 2) routing to appropriate SQL generation method; and 3) three-tiered SQL generation pipelines ($G_B$: Basic, $G_M$: Intermediate, $G_A$: Advanced, with performance and cost increasing in this order).

**Formulation.** Formally, when deploying an LLM $M$, the schema linking in Phase I can be denoted as $\mathcal{S}'_{\mathcal{D}} = M(\mathcal{S}_{\mathcal{D}}, \mathcal{Q})$, where $\mathcal{S}'_{\mathcal{D}}$ represents the filtered schema.

### 3.3 Phase II: Routing

**Overview.** Within the framework, the router is strategically positioned in Phase II, following the schema linking phase, as schema linking constitutes a standard preprocessing in Text-to-SQL. More importantly, the filtered schema $\mathcal{S}'_{\mathcal{D}}$ provides valuable structural information that strongly correlates with query complexity. For instance, heuristically, a query with a linked schema involving only a single table and two columns (*e.g.*, "List all customers' names from California") can be less complex than one requiring joins across multiple tables with numerous columns (*e.g.*, "Find the average order value of premium customers who made purchases in both January and February"). Additionally, the natural language query itself contains important complexity cues, particularly regarding functions or calculations potentially required in the SQL generation (*e.g.*, COUNT or STRFTIME) (Li et al., 2025). Consequently, in Phase II, the router is well-positioned to estimate the complexity of the query at hand by leveraging both the natural language query $\mathcal{Q}$ and its corresponding filtered schema $\mathcal{S}'_{\mathcal{D}}$. We discuss various router implementations in Section 4.1.

**Formulation.** Let $\mathbb{G}$ denote the set of available SQL generation pipelines. The router $\mathcal{R}$, is defined as a function targeted to map a natural language query $\mathcal{Q}$, and its corresponding filtered schema $\mathcal{S}'_{\mathcal{D}}$, to an optimal SQL generation pipeline $G_* \in \mathbb{G}$ which is the most efficient method capable for the query. Specifically, the router $\mathcal{R}$ utilizes $\mathcal{Q}$ and $\mathcal{S}'_{\mathcal{D}}$ to estimate query difficulty in its routing decision. Formally, the routing process is denoted as $\mathcal{R} : \mathcal{Q} \times \mathcal{S}'_{\mathcal{D}} \to \mathbb{G}$, such that $\mathcal{R}(\mathcal{Q}, \mathcal{S}'_{\mathcal{D}}) = G_*$.

### 3.4 Phase III: SQL Generation Pipelines

**Overview.** The three-tiered SQL generation pipelines in EllieSQL tackle the fundamental trade-off between performance and computational cost. We introduce a hierarchical set of three-tiered SQL generation pipelines—Basic ($G_B$), Intermediate ($G_M$), and Advanced ($G_A$)—each offering progressively better performance at correspondingly higher token consumption. This tiered structure enables our router to assign queries to the most cost-efficient pipeline capable of handling a given query's complexity.

**Formulation.** Formally, each SQL generation pipeline $G_i \in \mathbb{G} = \{G_B, G_M, G_A\}$ takes as input a natural language query $\mathcal{Q}$ and its schema linking result $\mathcal{S}'_{\mathcal{D}}$, producing a SQL query $y = G_i(\mathcal{Q}, \mathcal{S}'_{\mathcal{D}})$. After the router assigns the generation method $G_*$ in Phase II, the SQL generation process in Phase III can be formulated as $y = G_*(\mathcal{Q}, \mathcal{S}'_{\mathcal{D}})$. We discuss the implementation of three-tiered SQL generation pipelines in Section 4.2.

### 3.5 Metrics

**EX.** Execution Accuracy (EX) (Yu et al., 2018) evaluates the performance of the Text-to-SQL system by comparing whether the execution result sets of the gold SQL queries and the predicted SQL queries are identical:

$$EX = \frac{\sum_{i=1}^{N} \mathbb{1}(V_i = \hat{V}_i)}{N}, \tag{1}$$

where $V_i$ and $\hat{V}_i$ represent the execution result sets of the $i$-th predicted and gold SQL query, and $N$ is the total number of examples. Let the $EX$ of basic ($G_B$), intermediate ($G_M$) and advanced ($G_A$) SQL generation pipelines be $EX_B$, $EX_M$ and $EX_A$, respectively.

**PGR.** Following Ong et al. (2025), we use Performance Gap Recovered (PGR) to evaluate routers' performance. This metric captures how much of the performance difference between the weak and strong methods is recovered by the router $\mathcal{R}$:

$$PGR_{\mathcal{R}} = \frac{EX_{\mathcal{R}} - EX_B}{EX_A - EX_B}. \tag{2}$$

**TEP.** Inspired by the concept of elasticity in economics (Marshall, 2013), which measures how responsive one variable is to changes in another, we define Token Elasticity of Performance (TEP) as a metric to evaluate how sensitively the performance of a SQL generation pipeline, denoted G, responds to changes in token investment. Token consumption for a method is defined as $T = T_{\text{in}} + \mu T_{\text{out}}$, where $T_{\text{in}}$ represents consumed prompt tokens (input), $T_{\text{out}}$ is the completion tokens (output), and $\mu$ is a multiplier factor for completion tokens. The average token consumption is then computed as $\overline{T} = T/N$, where $N$ is the number of samples. TEP is formally expressed as:

$$TEP_G = \frac{\% \text{ change in performance}}{\% \text{ change in average token consumption}} = \frac{\Delta EX_G/EX_B}{\Delta \overline{T}_G/\overline{T}_B}, \tag{3}$$

where $\Delta EX_G = EX_G - EX_B$ represents the performance improvement of the tested SQL generation method G over a baseline method $G_B$, measured in terms of $EX$. Similarly, $\Delta \overline{T}_G = \overline{T}_G - \overline{T}_B$ denotes the additional average token consumption of method G relative to the baseline. A higher TEP value indicates greater efficiency in translating token investment into performance gains, reflecting better cost-effectiveness.

## 4 Implementation Details

### 4.1 Router Implementation

We investigate various router implementations across three categories for experiments: classification-based routers, cascading routers, and preference learning-based routers.

**Classification-based Routers** directly predict the most suitable SQL generation pipeline from the set of available methods G by treating routing as a multi-class classification problem. We explore two types of classification approaches: k-nearest neighbors (KNN) and supervised fine-tuning (SFT) lightweight language models as classifiers.

- *KNN*. The KNN router represents a simple heuristic approach: Queries involving more tables and columns are likely to be more complex, potentially requiring more sophisticated generation pipelines. Using a training set of queries with known optimal SQL generation pipeline allocations, it extracts feature vectors $v = (|\mathcal{T}|, |\mathcal{C}|)$ representing the number of tables and columns in $\mathcal{S}'_{\mathcal{D}}$. For a new query, it 1) Extracts $v$, 2) Finds $k$ nearest neighbors in the training set, and 3) Assigns the most common pipeline among neighbors.

- *SFT*. We fine-tune two lightweight language models as classifiers: 1) RoBERTa-base, an encoder-only model; 2) Qwen2.5-0.5B, with an added linear classification head that maps its hidden states to probabilities of SQL generation pipelines for multi-class classification.

**Cascading Routers** implement a sequential decision-making process using a series of binary classifiers. Each generation method has a corresponding classifier that determines if it can handle the current query. Different from Varshney & Baral (2022); Chen et al. (2023), our cascading routers still predictively route queries without performing SQL generation. The cascading routing follows a waterfall pattern:

$$\mathcal{R}_{\text{casc}}(\mathcal{Q}, \mathcal{S}'_{\mathcal{D}}) = \begin{cases} G_1 & \text{if } \mathcal{B}_1(\mathcal{Q}, \mathcal{S}'_{\mathcal{D}}) = 1 \\ G_i & \text{if } \left( \bigwedge_{j=1}^{i-1} \mathcal{B}_j(\mathcal{Q}, \mathcal{S}'_{\mathcal{D}}) = 0 \right) \wedge \mathcal{B}_i(\mathcal{Q}, \mathcal{S}'_{\mathcal{D}}) = 1, i \in \{2, \ldots, n-1\} \\ G_n & \text{otherwise} \end{cases} , \quad (4)$$

where $\mathcal{B}_i$ is the binary classifier for method $G_i$. We implement these classifiers by fine-tuning RoBERTa-base and Qwen2.5-0.5B models. Compared with multi-class classification, the cascading approach offers better extensibility (adding methods requires only one new classifier without affecting existing ones) and eases data collection.

**Preference Learning-based Routers** approach the routing task as learning relative preferences between generation methods rather than making hard classifications. We construct preference pairs $(G_i \succ G_j | \mathcal{Q}, \mathcal{S}'_{\mathcal{D}})$ and fine-tune Qwen2.5-0.5B with two preference learning techniques: Pairwise Ranking and Direct Preference Optimization (DPO).

- *Pairwise Ranking* is used to train the model to distinguish between better and worse generation methods using a margin-based ranking loss (Joachims, 2002):

$$\mathcal{L}_{\text{rank}} = \max(0, \lambda - s(G_i | \mathcal{Q}, \mathcal{S}'_{\mathcal{D}}) + s(G_j | \mathcal{Q}, \mathcal{S}'_{\mathcal{D}})), \quad (5)$$

  where $s(G | \mathcal{Q}, \mathcal{S}'_{\mathcal{D}})$ is the logit assigned to $G$, and $\lambda$ is the margin hyperparameter.

- *DPO* directly optimizes the preference model by maximizing the probability of preferred generations over less preferred ones (Rafailov et al., 2023) using:

$$\mathcal{L}_{\text{DPO}} = -\mathbb{E}_{(G_i \succ G_j, \mathcal{Q}, \mathcal{S}'_{\mathcal{D}})} \left[ \log \sigma(\beta \log \frac{\pi_\theta(G_i | \mathcal{Q}, \mathcal{S}'_{\mathcal{D}})}{\pi_{\text{ref}}(G_i | \mathcal{Q}, \mathcal{S}'_{\mathcal{D}})} - \beta \log \frac{\pi_\theta(G_j | \mathcal{Q}, \mathcal{S}'_{\mathcal{D}})}{\pi_{\text{ref}}(G_j | \mathcal{Q}, \mathcal{S}'_{\mathcal{D}})}) \right], \quad (6)$$

  where $\pi_\theta$ is the policy model being trained, and $\pi_{\text{ref}}$ is a reference model that remains unchanged during the training process. The $\beta$ is a temperature parameter controlling the divergence between $\pi_\theta$ and $\pi_{\text{ref}}$.

## 4.2 Three-tiered SQL Generation Pipelines Implementation

We introduce the implementation of our three-tiered SQL generation pipelines in Phase III as follows. All of the used prompts are shown in Appendix Section A.3.

- *Basic SQL Generation Pipeline* ($G_B$). This tier adopts a direct input-output approach, generating SQL queries solely based on the given natural language query and schema linking results, without any intermediate reasoning steps. Although it is less effective for complex queries, it can still handle straightforward cases. As the simplest and most lightweight method, it has the lowest token consumption.

- *Intermediate SQL Generation Pipeline* ($G_M$). To improve the handling of complex queries, this tier follows CHASE-SQL (Pourreza et al., 2025) and deploys a Divide-and-Conquer Chain-of-Thought strategy. Specifically, as illustrated by Phase III-$G_M$ in Figure 1, the original query is decomposed into sub-questions, which are solved independently and then combined into a complete solution. A final refinement step is applied to correct any issues in the generated SQL, such as non-executable queries or queries yielding empty results. This method is reported to be more effective for handling complex queries but comes with increased token consumption per example.

- *Advanced SQL Generation Pipeline* ($G_A$). Building on the intermediate tier, this approach further integrates the online synthesis mechanism adopted from CHASE-SQL (Pourreza et al., 2025) to enhance SQL generation. In addition to the decomposition and refinement steps in $G_M$, online synthesis is applied to each subtask. This mechanism dynamically

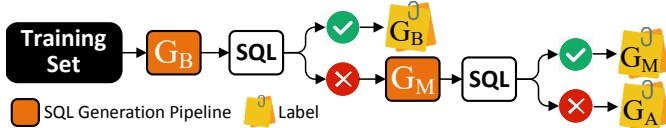

Figure 2: Training data construction for routers

| | $G_B$ | $G_M$ | $G_A$ | Total |
|---|---|---|---|---|
| | 4770 | 776 | 3476 | 9012 |

Table 1: Distribution of assigned labels in training set

synthesizes relevant examples based on the given schema as shown in Figure 1 Phase III-$G_A$, providing specific few-shot demonstrations tailored to the test query. By constructing demonstrations using pertinent tables and columns, this approach can gain a better understanding of the underlying data schema, making $G_A$ the most powerful—but also the most computationally expensive—base method in our study.

# 5 Experiments

## 5.1 Experimental Setup

**Dataset.** Bird (Li et al., 2024b) provides cross-domain Text-to-SQL tasks, which are diverse and proportionate in terms of query complexity. Therefore, we utilize the Bird development set with 1534 (NL, SQL) pairs on 11 databases for evaluation and the Bird training set with 9012 valid (NL, SQL) pairs on 69 databases for fine-tuning. Importantly, the training and development sets in Bird are constructed based on entirely separate databases without any overlap, ensuring that the router is challenged to generalize across different databases during evaluation. Moreover, to evaluate the generalizability of EllieSQL, we assess its out-of-distribution (OOD) performance using the Spider-dev, Spider-Realistic-dev (Deng et al., 2021), and Spider-test (Yu et al., 2018) datasets.

To train our routers, we construct the training data using a waterfall approach, as shown in Figure 2. Each query is first attempted with the basic SQL generation pipeline ($G_B$); if successful, it receives a $G_B$ label. Failed queries cascaded to the next-tier intermediate pipeline ($G_M$) and then to the advanced pipeline ($G_A$) if necessary. In this way, each query is assigned the label with the most efficient capable SQL generation pipeline. Table 1 shows the label distribution in the training set. For preference learning, we derive preference pairs from the label, such as $G_B \succ G_M$ and $G_B \succ G_A$ for a query assigned with label $G_B$.

**Training and Evaluation.** As mentioned in Section 3.3, RoBERTa-base and Qwen2.5-0.5B are used as base models to train different routers. For brevity, hereafter, we refer to them as RoBERTa and Qwen, respectively. We fine-tune our routers using the LoRA (Hu et al., 2022) on four RTX 4090 GPUs with a learning rate of 1e-4. For evaluation, to allow a fair and controlled assessment of the performance and cost-efficiency inherent in each SQL generation pipeline, we consistently deploy `gpt-4o-mini-2024-07-18` as the backbone model across three tiers of SQL generation pipelines in Phase III, where $\mu = 4$ and all temperatures are set to 0.

**Experiment Objective.** The primary objective of our experiments is to validate whether our complexity-aware routing framework for Text-to-SQL can maintain performance comparable to consistently using the most advanced pipeline while significantly reducing token consumption. Therefore, our analysis *focuses on relative performance differences and comparisons* between different methods, rather than *absolute values*.

## 5.2 Experimental Results and Analysis

### 5.2.1 Performance

As illustrated in Table 2, our experimental results reveal a clear performance progression across base methods on the Bird dev, with Advanced ($G_A$) outperforming Intermediate ($G_M$) and Basic ($G_B$) SQL generation pipelines, particularly on hard cases. Among routing approaches, Qwen DPO and RoBERTa Cascading stand out as the top routers with no performance sacrifice—even slightly outperforming the standalone $G_A$. The Qwen DPO router demonstrates best performance in challenging queries among routers though still slightly weaker then consistently using $G_A$. Meanwhile, most router implementations

| Category | Method | EX(%) Performance | | | | Cost-efficiency | | | |
| --- | --- | --- | --- | --- | --- | --- | --- | --- | --- |
| | | Total↑ | Sim. | Mod. | Chal. | $\overline{\text{T}}$↓ | PGR↑ | TEP↑$(\times 10^{-2})$ | Time↓ |
| Base | Basic ($G_B$) | 51.83 | 59.78 | 42.03 | 32.41 | 695.55 | - | - | **114** |
| | Intermediate ($G_M$) | 54.17 | 61.62 | **44.40** | 37.93 | 11792.16 | - | 0.283 | 609 |
| | Advanced ($G_A$) | **55.02** | **62.07** | 43.53 | **42.76** | 13002.91 | - | 0.348 | 653 |
| Routing | KNN | 52.93 | 60.97 | 42.24 | 35.86 | 3615.67 | 0.345 | 0.506 | **326** |
| | RoBERTa SFT | 54.30 | 61.95 | 44.83 | 35.86 | 6244.72 | 0.774 | 0.597 | 416 |
| | Qwen SFT | 54.30 | 61.95 | 44.18 | 37.93 | 7220.68 | 0.774 | 0.508 | 402 |
| | RoBERTa Casc. | 55.15 | **62.92** | 45.04 | 37.93 | 8211.00 | 1.040 | 0.593 | 467 |
| | Qwen Casc. | 54.50 | 62.38 | 44.18 | 37.24 | 8481.66 | 0.836 | 0.460 | 644 |
| | Qwen Pair. Rank | 53.72 | 60.22 | 45.47 | 38.62 | 6476.39 | 0.592 | 0.439 | 332 |
| | Qwen DPO | **55.41** | 62.38 | **45.91** | **41.38** | 7641.51 | **1.122** | **0.692** | 396 |

Table 2: Execution Accuracy (EX) with breakdowns across Simple (Sim.), Moderate (Mod.), and Challenging (Chal.) examples, Average Token Consumption per example ($\overline{\text{T}}$), Performance Gap Recovered (PGR) by Router, Token Elasticity of Performance (TEP), and inference time duration in seconds on Bird dev. The **bold** and underlined results are the best and second best in each category.

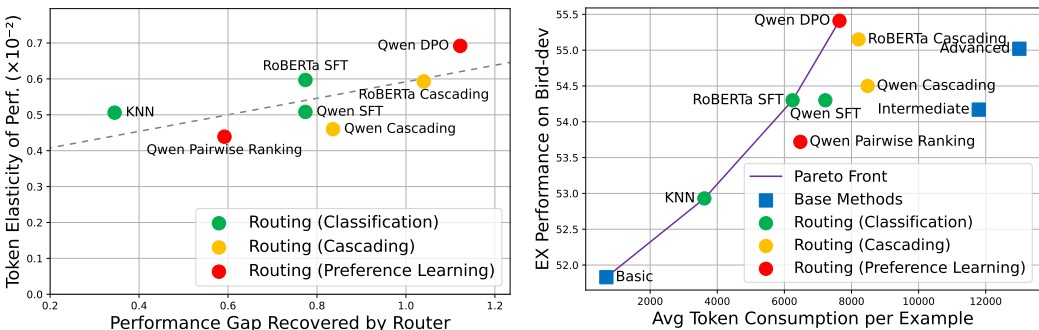

Figure 3: Performance Gap Recovered (PGR) by Router vs. Token Elasticity of Performance (TEP) on Bird dev.

Figure 4: EX performance vs. average token consumption per example ($\overline{\text{T}}$) on the Bird dev, and the Pareto Frontier.

showcase competitive performance, with approaches like RoBERTa SFT, Qwen SFT, and Qwen Cascading achieving better overall execution accuracy than consistently using $G_B$. Even when these approaches perform slightly below consistently using $G_A$, the performance degradation remains acceptable. These results exhibit the feasibility and potential of complexity-aware routing in Text-to-SQL, confirming that with appropriate routing strategies, we can match the performance of consistently using the most advanced method, even without compromising performance.

Additionally, as outlined in Section 5.1, the training and development sets are partitioned with no overlapping databases, which challenges routers to generalize effectively across distinct databases, thus their strong performance also further presents transferability and adaptability across diverse databases.

### 5.2.2 Cost-efficiency

The right side in Table 2 presents the average token consumption in SQL generation, Performance Gap Recovered (PGR), and Token Elasticity of Performance (TEP) across different methods. The PGR assesses the effectiveness of routers (Ong et al., 2025), while TEP measures the responsiveness of performance gains relative to token investments.

Base methods exhibit a clear trade-off between performance and cost, with more powerful SQL generation pipelines requiring significantly more tokens than Basic methods to achieve the performance improvements noted in the middle part of Table 2, resulting in lower TEP. In contrast, our routing approaches have proven effective by demonstrating greater efficiency in converting token investment into performance gains, with all router implementations showing improved TEP. With the best-performing router, Qwen DPO, SQL generation consumes over 40% fewer tokens than $G_A$ without sacrificing performance. This doubles the

| Category | Method | Spider-dev | | | Spider-Realistic-dev | | | Spider-test | | |
|---|---|---|---|---|---|---|---|---|---|---|
| | | EX(%)↑ | $\overline{T}$↓ | TEP↑$_{(\times 10^{-2})}$ | EX(%)↑ | $\overline{T}$↓ | TEP↑$_{(\times 10^{-2})}$ | EX(%)↑ | $\overline{T}$↓ | TEP↑$_{(\times 10^{-2})}$ |
| Base | Basic ($G_B$) | 74.27 | 363.40 | - | 73.23 | 377.95 | - | 74.29 | 376.14 | - |
| | Intermediate ($G_M$) | 75.73 | 2432.13 | 0.345 | 74.80 | 2520.67 | 0.378 | 76.90 | 2422.84 | 0.645 |
| | Advanced ($G_A$) | **76.60** | 3826.14 | 0.356 | **75.39** | 3845.75 | 0.321 | **77.69** | 3584.01 | 0.537 |
| Routing | KNN | 75.05 | 1033.28 | 0.570 | 72.64 | 1171.93 | -0.383 | 75.08 | 974.26 | 0.669 |
| | RoBERTa SFT | 75.92 | 1222.21 | **0.940** | 73.43 | 1128.54 | 0.138 | 75.78 | 1130.04 | 1.001 |
| | Qwen SFT | 75.63 | 2014.60 | 0.403 | 75.39 | 2064.06 | 0.661 | 75.55 | 2196.56 | 0.350 |
| | RoBERTa Casc. | 76.21 | 1516.21 | 0.823 | 74.21 | 1593.84 | 0.416 | 76.90 | 1539.21 | 1.136 |
| | Qwen Casc. | 75.53 | 3410.96 | 0.202 | **75.98** | 3526.75 | 0.451 | 77.04 | 3296.22 | 0.477 |
| | Qwen Pair. Rank | 74.76 | 2334.71 | 0.122 | 74.61 | 2379.99 | 0.356 | 75.08 | 2245.64 | 0.214 |
| | Qwen DPO | 76.31 | 1906.79 | 0.647 | 75.79 | 1721.43 | **0.983** | 77.97 | 1850.88 | **1.263** |

Table 3: Experimental results on out-of-distribution (OOD) datasets. The **bold** and underlined results are the best and second best in each category.

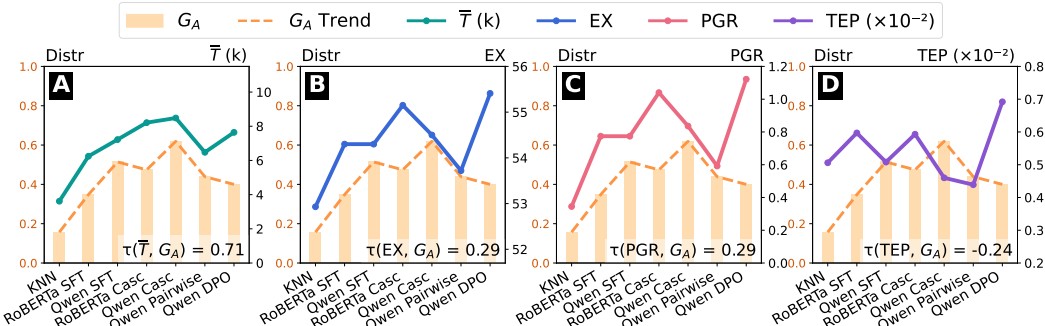

Figure 5: Comparison of $G_A$ allocation in routing with other metrics.

TEP, making it twice as responsive to token consumption in terms of performance gains and achieving the best cost-efficiency. Although the RoBERTa SFT-based router experiences a slight performance loss, it also demonstrates competitive cost-efficiency. Cascading routers are competitive in EX and routing performance but are slightly inferior in TEP compared to SFT-based routers due to higher token costs.

Figure 3 illustrates the relationship between routers' PGR performance and TEP for SQL generation, revealing an overall positive correlation. This implies that superior routing performance tends to demonstrate greater token elasticity during SQL generation. Figure 4 depicts the EX performance and token cost with a *Pareto Frontier* (purple line), highlighting the cost-efficiency advantages of routers based on RoBERTa SFT and Qwen DPO. Nevertheless, even though the RoBERTa Cascading router is dominated by Qwen DPO on the Pareto Frontier, it remains a practical option in many scenarios due to its superior extensibility, as adding new methods requires only one new classifier without affecting existing ones, which can ease the data preparation and router training.

EllieSQL's efficiency extends to processing speed. Notably, under uniform settings, experiments on Bird dev also demonstrate that Qwen DPO router reduce total time duration by over 40% compared to the consistent deployment of $G_A$. However, the cascading router design, while offering superior extensibility for future integrations, inherently demands more processing time. This presents a trade-off between immediate computational efficiency and the system's long-term architectural scalability.

### 5.2.3 Generalizability on Out-of-Distribution (OOD) Datasets

To assess the generalizability of EllieSQL, we further evaluate on several OOD datasets: Spider-dev, Spider-Realistic, and the Spider-test set. Note that routers are trained exclusively on the Bird training set and then directly deployed on these OOD datasets without any fine-tuning on the Spider training set.

As presented in Table 3, across all three OOD benchmarks, our proposed EllieSQL, particularly with Qwen DPO and cascading routers, significantly reduces token costs while maintaining high performance. Specifically, on the Spider-test, EllieSQL deployed with Qwen DPO router not only surpasses $G_A$ in performance but does so with a 48% token reduction, yielding a superior TEP. Likewise, on Spider-Realistic-dev, EllieSQL with multi-

ple routers, including Qwen DPO, Qwen SFT and Qwen Cascading, achieves even better performance than standalone $G_A$ while reducing token consumption by 55%, 46%, 8%, respectively, doubling or even tripling the TEP metric. The strong OOD performance validates that EllieSQL generalizes effectively to unseen datasets, confirming its robustness and efficiency for diverse Text-to-SQL tasks.

### 5.2.4 *Strategic Routing vs. Advanced Method Selection.*

Figure 5 shows the proportion of $G_A$ assigned in routing across different router implementations. We compare the usage frequency of the most powerful and costly SQL generation pipeline $G_A$ with other metrics, to analyze their relationship and investigate whether performance improvements across different routers simply come from increased selection of advanced methods or indeed stem from effective routing decisions.

As shown in Figure 5, obviously, higher allocation of $G_A$ is positively correlated with increased average token consumption. However, the EX performance exhibits a different pattern. Although the Qwen Cascading router assigns $G_A$ more frequently than any other router and incurs the highest token consumption, it does not rank at the top in terms of EX performance, resulting in a low TEP metric and indicating inefficiency. In contrast, RoBERTa Cascading and Qwen DPO routers select less $G_A$ than Qwen Cascading and Qwen SFT routers, yet they yield top performance and lead in TEP, proving their effectiveness, particularly the Qwen DPO router. This substantiates that performance gains are indeed driven by the effectiveness of routing strategies rather than merely deploying sophisticated methods more frequently.

## 6 Limitations and Future Directions

EllieSQL showcases better cost-efficiency via complexity-aware routing, yet limitations persist. First, labeled training data introduces an upfront cost for router implementation, although this expense becomes increasingly amortized with wider system adoption and EllieSQL presents great database transferability. Time and funding restricted us to three base methods and router categories, leaving room to explore additional SQL generation approaches (*e.g.,* MCTS-enhanced reasoning) and more advanced routing techniques.

Another limitation is the router's dependency on the upstream schema linking stage, where errors could propagate and may affect subsequent routing. We consider this a pragmatic design and validate the robustness of schema linking in our experiments, which presents a 93% high recall while reducing the column by over 90% on average. More advanced schema linking methods could further strengthen downstream routing performance.

Also, introducing new SQL generation pipelines requires retraining new routers, which introduces incremental costs associated with data annotation and training. Future work could target data-efficient router training even training-free routing, while pursuing more scalable and generalizable routers for more sustainable and practical Text-to-SQL solutions.

## 7 Conclusion

In this paper, we propose EllieSQL to tackle the "elephant" in current leaderboard-driven Text-to-SQL research: the unsustainable computational costs. By dynamically assigning queries to different tiered SQL generation pipelines based on estimated complexity, EllieSQL seeks to balance between performance and efficiency. We also introduce Token Elasticity of Performance (TEP), an economic-inspired metric that evaluates the responsiveness of performance gains relative to token investments in SQL generation. Experiments demonstrate EllieSQL 's effectiveness: various router implementations significantly reduce token consumption while preserving competitive performance, even showing no performance compromise compared to consistently using the most advanced methods. Our findings establish the feasibility and potential of complexity-aware routing in Text-to-SQL, paving the way for further exploration of more advanced routing strategies. Ultimately, we hope EllieSQL can encourage the community to balance resource efficiency with performance on leaderboards, fostering efforts toward sustainable Text-to-SQL solutions that are both high-performing and cost-efficient for real-world adoption.

## Acknowledgments

This paper is supported by NSF of China (62402409), Guangdong provincial project 2023CX10X008, Guangdong Basic and Applied Basic Research Foundation (2023A1515110545), Guangzhou Basic and Applied Basic Research Foundation (2025A04J3935), HKUST(GZ) Red Bird M.Phil. Program, and Guangzhou-HKUST(GZ) Joint Funding Program (2025A03J3714).

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

# A  Appendix

## A.1  Model Transferability

As mentioned in Section 5.1, given that the training set and development set in the Bird dataset are constructed from distinct databases, routers are challenged to generalize across different databases. Their performance, as discussed in Section 5.2, highlights the transferability of EllieSQL across diverse databases. To further investigate the model transferability of routers within the EllieSQL, we experiment by replacing the backbone model in the SQL generation task with `claude-3-haiku-20240307` and much larger `gpt-4o-2024-08-06` respectively, without re-training any of the existing routers. Due to financial constraints, we examine three representative routers in our experiments.

As reported in Table 4, the original routers demonstrate effectiveness even after the backbone model substitution, showcasing transferability. These routers significantly enhance the responsiveness of performance gains relative to token investment, while maintaining competitive performance levels.

Specifically, when using `claude-3-haiku-20240307` as the backbone LLM, the RoBERTa Cascading and Qwen SFT routers are effective in this transferability analysis. They achieve overall performance close to the standalone $G_A$ and reduce token costs by 39.5% and 45.8%, respectively. This leads to an improvement in Token Efficiency Performance (TEP) by 29.8% and 34.4%. Furthermore, when `gpt-4o-2024-08-06` is deployed as the backbone in EllieSQL, all three examined routers also demonstrate high effectiveness. In particular, the RoBERTa Cascading router and the Qwen DPO router reduce token consumption by 41% and 44%, respectively, without any performance sacrifice, which significantly improves the TEP metric, increasing it by more than 2.5 times. These findings highlight the adaptability and efficiency of the proposed EllieSQL framework.

| Category | Method | claude-3-haiku-20240307 | | | gpt-4o-2024-08-06 | | |
|---|---|---|---|---|---|---|---|
| | | EX(%)↑ | $\overline{T}$↓ | TEP↑$(\times 10^{-2})$ | EX(%)↑ | $\overline{T}$↓ | TEP↑$(\times 10^{-2})$ |
| Base | Basic ($G_B$) | 51.17 | 736.91 | - | 56.13 | 768.03 | - |
| | Intermediate ($G_M$) | 51.76 | 12153.93 | 0.074 | 57.82 | 9357.91 | 0.269 |
| | Advanced ($G_A$) | 54.37 | 13444.51 | 0.363 | 58.34 | 10360.15 | 0.315 |
| Routing | RoBERTa SFT | 52.48 | 6094.96 | 0.352 | 58.15 | 4426.60 | 0.755 |
| | RoBERTa Casc. | 53.59 | 8136.75 | 0.471 | 58.87 | 6073.55 | 0.706 |
| | Qwen DPO | 53.26 | 7792.88 | 0.427 | 59.39 | 5832.57 | 0.881 |

Table 4: Performance of EllieSQL with `claude-3-haiku-20240307` and `gpt-4o-2024-08-06` as backbone model on Bird dev dataset. The **bold** and underlined results are the best and second best in each category, respectively.

## A.2  Comparison with Oracle Labels

We also deploy the process in Figure 2 to the Bird development set to obtain Oracle labels in evaluation. Note that Oracle labels are the most efficient SQL generation pipeline capable for each query, which are guidance labels instead of absolute ground truth. Figure 7 shows the distribution of Oracle labels on the Bird development set.

We measure the agreement between the router's predictions and the Oracle labels, and compare the agreement with EX performance, as illustrated in Figure 6-A. Our analysis reveals a weak monotonic relationship between agreement with Oracle labels and EX performance (with only negligible Kendall's $\tau$). Regarding disagreements with Oracle, directing a complex query to a simple method is *more detrimental* than directing a simple query to a complex method. While the latter may just incur some token overhead, the former can significantly increase the risk of errors.

To formalize this, we can use a disagreement matrix $C$ to represent the router's predictions and Oracle labels. Disagreements above the diagonal (simple queries routed to complex pipelines) are less severe, whereas those below the diagonal (complex queries routed to

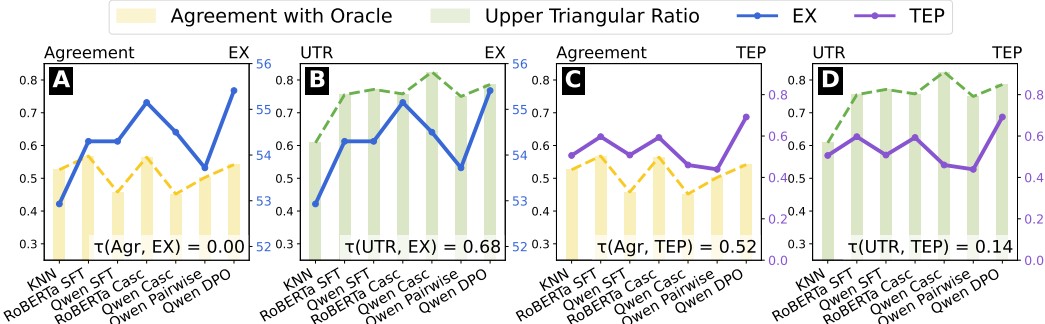

Figure 6: Comparison of agreement with Oracle and Upper Triangular Ratio (UTR) in routing with EX and TEP.

simple pipelines) are more detrimental, as discussed above. Thus, the Upper Triangular Ratio (UTR) can reflect the "reliability" of a router, since high UTR indicates a reduced risk of under-processing complex queries. Formally,

$$UTR = \frac{\sum_{i \le j} C_{ij}}{\sum_{i,j} C_{ij}}, \tag{7}$$

where $C_{ij}$ is the number of queries in cell $(i, j)$. While higher UTR correlates with reduced misdirection of complex queries to simple methods (thereby improving reliability), it is insufficient as a standalone metric for effectiveness of routing. For instance, assigning all queries to the most complex pipeline would yield a full UTR of 1, yet such a strategy constitutes a trivial and ineffective routing mechanism.

Figures 6-A/C demonstrate that agreement with Oracle exhibits a partial correlation with TEP but fails to reflect EX performance. In contrast, Figures 6-B/D reveal that while UTR achieves a moderate correlation with EX (Kendall's $\tau = 0.68$), it suffers from significant limitations in assessing routing effectiveness. Specifically, RoBERTA SFT and Qwen Cascading routers present the highest agreement and UTR, respectively, but they do not demonstrate top performance either in overall EX or TEP. Therefore, naive agreement-ratio-based comparisons are inadequate for evaluating routing performance or cost-efficiency, motivating the introduction of the PGR metric (Ong et al., 2025) and TEP introduced in Section 3.5.

Figure 8 presents the disagreement matrix of the Qwen DPO router. While it is more effective at discerning tasks suitable for allocation to $G_M$ and $G_A$, the matrix suggests that distinguishing $G_M$ poses a greater challenge for the router, likely due to the limited number of examples (Table 1 and Figure 7) and relatively narrow performance gap between $G_M$ and $G_A$. Although our best-performing router, Qwen-DPO, achieves a decent UTR of 0.79, there still remains notable room for further reducing below-diagonal disagreements in future work to enhance reliability. This suggests considerable potential for enhancing Text-to-SQL routers in future studies, thereby improving both the overall performance and cost-efficiency of the Text-to-SQL system.

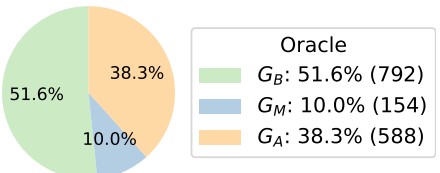

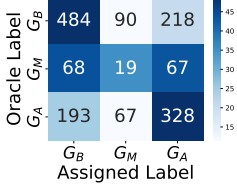

Figure 7: Distribution of Oracle labels in evaluation on Bird dev

Figure 8: Disagreement matrix between Oracle labels and Qwen DPO Router Predictions

### A.3 Prompt Template

#### A.3.1 Prompt for Schema Linking

---

**Schema Linking Prompt**

You are a smart and responsible SQLite SQL expert. Assist in identifying the database tables and columns involved in natural language queries.

### Instruction:
Your task is to analyze the provided database schema, comprehend the posed question, and leverage the hint to identify which tables are needed to generate a SQL query for answering the question. The returned JSON format must strictly adhere to the following specifications:

```
{
    "tables": [
        {
            "table": "table name",
            "columns": ["relevant column 1", "relevant column 2", ...]
        },
        ...
    ]
}
```

Each relevant column must belong to its respective table, and the output JSON object must be wrapped in a code block using ```json```. Please note that each table and column comes with detailed description information and example values for reference.

### Database schema:
{schema_str}

### User question:
{query}

### Hint: {evidence}

---

#### A.3.2 Prompt for Basic SQL Generation Pipeline

---

**Basic SQL Generation Pipeline**

You are a intelligent and responsible SQLite expert.

### Instruction:
You need to read the database schema to generate SQL query for the user question. The outputted SQL must be surrounded by ```sql``` code block.

### Database Schema:
{schema}

### Hint:
{evidence}

### User Question:
{query}

The outputted SQL must be surrounded by ```sql``` code block.

---

### A.3.3  Prompt for Divide-and-Conquer Chain-of-Thought

---

**Divide Prompt**

You are a smart and responsible SQLite SQL expert. Given a database schema and a question, users want to know the corresponding SQL query. Your task is to understand the database schema and question, and decompose the question into sub-questions so user can better understand it. Each sub-question is enclosed in <<>>. Here is an example for reference:

### Example:

## Given the database schema:
{example_database_schema}

## Question:
{example_question}

## Decompose the Question into sub-questions, each sub-question is enclosed in <<>>:
Sub-question 1: <<{sub question 1}>>
Sub-question 2: <<{sub question 2}>>
Sub-question 3: <<{sub question 3}>>

### Your task: decompose the question into sub-questions.

## Given the database schema:
{schema}

## Question:
{query}

## Hint:
{evidence}

## Decompose the Question into sub-questions, each sub-question is enclosed in <<>>:

---

**Conquer Prompt**

You are a smart and responsible SQLite SQL expert. Given a database schema and a question, your tasks are:

1. Parse user questions: Use natural language processing (NLP) techniques to parse user questions and extract query requirements and conditions.

2. Analyze database schema: Based on the database schema, understand the fields and relationships of the table, and build the basic framework of the SQL query.

3. Check sample data: Analyze the data characteristics based on the first three rows of the table values to help determine how to construct query conditions and filter results.

4. Generate SQL query: Based on user questions, query requirements and conditions, database schema, and sample data, build a complete SQL query.

5. Verification and optimization: Check whether the generated SQL query is logical and optimize it if necessary.

### Database Schema:
{schema}

### Examples:
{examples}

### Question:
{query}

### Hint:
{evidence}

Please generate the corresponding SQL query. SQL must be surrounded by ```sql``` code block.

**Assemble Prompt**

You are a smart and responsible SQLite SQL expert. Given a database schema and a question, users want to know the corresponding SQL query.

### Instructions:
We have decomposed the main question into sub-questions, now your task is based on the SQL querys for corresponding sub-questions, assemble the final SQL for the main question:
1. Understand the database schema and the main question;
2. Read and analyze each sub-question and corresponding SQL query;
3. Analyze the relationship between sub-questions and the main question in order to assemble them properly;
4. Generate the final SQL for the main question and optimize it if needed.

### Database Schema:
{schema}

### Main question:
{query}

### Hint:
{evidence}

### Sub-questions and corresponding output, including SQL querys and explanation:
{subs}

Based on the SQL querys for corresponding sub-questions, generate the final SQL for the main question in the end of your response, SQL must be surrounded by ```sql``` code block.

### A.3.4  *Prompt for Online Synthesis Mechanism*

---

**Online Synthesis Mechanism Prompt**

### Instruction:
You are a SQLite SQL expert. Your job is to create {k} examples, where each example consists of a question and a SQL query to fetch the data for it. I want each example to look like this, question input and SQL output pairs:

### Example:

```
"Question": "What's the description of the series code SM.POP.TOTL for Aruba?
(Hints: Aruba is the name of the country where ShortName = 'Aruba')"
```

```
"SQL": "SELECT T2.Description FROM Country AS T1 INNER JOIN CountryNotes AS T2
ON T1.CountryCode = T2.Countrycode WHERE T1.ShortName = 'Aruba' AND
T2.Seriescode = 'SM.POP.TOTL'"
```

### Task:
You should generate examples that examine and showcase different aspects and relationships of the following table schemas, described in "Table creation statements". Understand the database tables and their relationships. Understand the columns and their types and meanings to construct interesting examples.

Generate a mixture of SQL examples that include:
• some simple SQL query examples without JOIN
• some SQL query examples with aggregates, like COUNT
• some simple SQL query examples with JOIN
• some complex SQL query examples with nested JOIN

## Database Schema:
{TARGET_DATABASE_SCHEMA}

Generate a total of {k} examples. Only output the examples ('question input' and 'SQL output' pairs), and each example can be separated by a new line.

---

