# OpenReview forum: "EllieSQL: Cost-Efficient Text-to-SQL with Complexity-Aware Routing"
_colmweb.org/COLM/2025/Conference — COLM 2025_

### Official Review · Reviewer_367x · 2025-05-10

**Rating:** 5
**Confidence:** 4
**Ethics Flag:** 1

**Summary:**

This paper presents a cost-efficient Text2SQL system with complexity-aware routing, specifically, this system consists of three stages: 1) schema linking stage to collect sufficient information (e.g. NLQ, filtered schema) for query complexity estimation; 2) routing stage to decide the query difficulties: basic, intermediate or advanced; 3) SQL generation pipelines to translate NLQ into the final SQL query. To study the effectiveness of different routers, this paper implemented three different methods: classification-based router, cascading router and preference learning-based router. By conducting experiments on BIRD benchmark, the proposed system shows comparable Text2SQL performance but much better token efficiency.

**Reasons To Accept:**

1. The proposed problem is very valuable and practically useful, not all NLQs share the same complexity, it is definitely needed to design a system to balance well between accuracy and inference cost.
2. Three router implementations shows novelty, the performance on the BIRD benchmark also demonstrates the effectiveness.

**Reasons To Reject:**

1. It's unclear about the router effectiveness on out-of-domain Text2SQL datasets, for example, we can evaluate the system on Spider related datasets (e.g. Spider, Spider-syn, Dr. Spider).
2. Some Text2SQL baselines (e.g. DIN-SQL [1] ) also have internal routing mechanism, for example, DIN-SQL [1] dynamically classifies query into EASY, NON-NESTED, or NESTED, then select different prompting mechanisms for SQL generation. It will be great if this paper can do a comparison with these works.
3. In addition to accuracy and token efficiency, in practical Text2SQL system, people also care about the latency, it's not clear about latency overhead for these router implementations.


[1] DIN-SQL: Decomposed In-Context Learning of Text-to-SQL with Self-Correction

---

> ### Author Response · Authors · 2025-06-02
> **Stage I Response (2/2)**
>
> > ### Question #2 on "Internal Routing"
>
> Thank you for highlighting this interesting aspect. In DIN-SQL, SQL queries are manually defined and categorized into four levels of complexity: Easy, Non-nested Complex, or Nested Complex. The LLM first classifies the query into one of these categories. Then, it is provided with a few hand-crafted, few-shot examples specific to that category to generate the final SQL. This approach aims to provide suitable examples to assist with different types of SQL generation, rather than primarily focusing on reducing token cost. In fact, DIN-SQL provides numerous examples for all task types (including Easy ones) and does not allocate computing resources based on complexity with the goal of cost reduction.
>
> In contrast, our routing mechanism aims to allocate a more lightweight approach for easier tasks while reserving computationally intensive methods for complex cases, thereby reducing token cost. Furthermore, the routers in our study are trained to learn complexity classification based on the results of schema linking, rather than relying on manually defined classification rules, such as the presence of "nested" queries. This is because the complexity of the text-to-SQL task is multifaceted and not solely related to the presence of nesting. For instance, SQL queries involving computational processes, merging multiple tables, or utilizing functions can significantly increase task complexity. Simple nesting, on its own, may not necessarily pose significant difficulties for advanced models. Therefore, we train routers to learn this classification rather than manually defining the rules.
>
> We will include a discussion of such a comparison in the related work section, thank you for this valuable suggestion.
>
> > ### Question #3 on Latency
>
> Following your suggestions, we tested the time duration on the Bird-dev dataset across different methods (with the same 64 max worker settings). The results are presented in the table below.
>
> Among the base methods, more complex SQL generation approaches yielded better performance but also incurred higher token costs and longer reasoning times to obtain the output SQL. The results also demonstrate the effectiveness of our routers: most routers not only reduced token costs but also decreased latency. Notably, with the Qwen DPO router, we achieved a reduction in token cost of over 40% and a decrease in total time duration of 40%, further highlighting its effectiveness. Moreover, although cascading routers have the advantage in extensibility, their cascading structure can lead to increased processing time, making them less competitive in terms of latency.
>
> | **Category** | **Method**         | **Bird dev EX** | **Tˉ** (avg token cost) | **TEP(x10^-2)** | **Time Duration (s)** |
> | ------------ | ------------------ | --------------- | ----------------------- | --------------- | --------------------- |
> | Base         | Basic              | 51.83           | 695.55                  | -               | 114                   |
> |              | Intermediate       | *54.17*         | 11792.16                | 0.283           | 609                   |
> |              | Advanced           | **55.02**       | 13002.91                | 0.348           | 653                   |
> | Routing      | KNN                | 52.93           | 3615.67                 | 0.506           | 326                   |
> |              | RoBERTa SFT        | 54.3            | 6244.72                 | *0.597*         | 416                   |
> |              | Qwen SFT           | 54.3            | 7220.68                 | 0.508           | 402                   |
> |              | RoBERTa Casc.      | *55.15*         | 8211                    | 0.593           | 467                   |
> |              | Qwen Casc.         | 54.5            | 8481.66                 | 0.46            | 644                   |
> |              | Qwen Pairwise Rank | 53.72           | 6476.39                 | 0.439           | **332**               |
> |              | Qwen DPO           | **55.41**       | 7641.51                 | **0.692**       | *396*                 |
>
> Thanks again for your insightful feedback. We hope our response helps clarify and address your concerns. If you have follow-up questions, please feel free to continue the discussion with us.

---

> > ### Comment · Reviewer_367x · 2025-06-05
> >
> > Thanks for providing detailed author responses!
> >
> > I am still concerned about the generalization capability of the proposed method, when I mention DIN-SQL, my point is that the internal routing can be finished by LLM (or orchestrator in an agent system), this internal routing is flexible to handle any Text2SQL systems (or we can treat each Text2SQL pipeline as a tool, the orchestrator can select the best one). However, in this paper, the router is trained with specific data and targeting on 3 Text2SQL pipelines, it's not easy to adapt to all future pipelines.
> >
> > Therefore I want to keep my current score unchanged.

---

> > ### Author Response · Authors · 2025-06-06
> >
> > Thank you for your continued engagement and valuable feedback!
> >
> >
> > > ### Clarification for the Main Focus of Our Study
> >
> > Firstly, we would like to clarify the **main focus of our study.** EllieSQL aims to highlight a critical yet often overlooked cost-efficiency limitation in current Text-to-SQL studies. Our goal is to significantly reduce token consumption without performance loss.
> >
> > To support this goal, we introduce a new metric TEP, and routing mechanism, and we investigate multiple router implementations to validate the effectiveness. Ultimately, we found that significant token reduction can be achieved without sacrificing performance, thereby validating the feasibility and effectiveness of EllieSQL.
> >
> > We hope to clarify our motivation and focus in this paper is **not to develop a state-of-the-art routing algorithm**, but to **demonstrate that effective routing can reduce token costs efficiently in text-to-SQL while maintaining the performance**. Future work can explore more advanced routers for improved scalability and efficiency—an interesting area for continued research.
> >
> > The **training-free routing based on LLMs** as you mentioned is essentially **another viable router implementation**, and it can be integrated into EllieSQL if desired by users.
> >
> > > ### On Generalization and Extensibility
> >
> > Meanwhile, we fully understand and appreciate your concerns on generalization capability to new pipelines in future. In our paper, that is the reason why we introduce **cascading routers**. For cascading structured routers, integrating a new method only requires adding a corresponding binary router, without retraining the original one. As previously discussed, this design improves flexibility, make it easier to integrate future pipelines, but does come at the cost of increased latency—there is no free lunch.
> >
> > > ### Additional Experiment
> >
> > To further address your concerns, following your insightful suggestion, we conducted an additional experiment where we use LLMs to perform routing (training-free).
> >
> > In this setting, we prompt the LLM (GPT-4o-mini) with a description of the tiered pipelines, the routing task, the user query, and the results of schema linking. The LLM then acts as a router to assign tasks into different pipelines. The results are shown below:
> >
> > | Method | Bird Dev EX | $\bar{T}$ (avg token cost) | TEP (×10⁻²) |
> > | --- | --- | --- | --- |
> > | LLM Routing | 53.78 | 11551.42 | 0.241 |
> >
> > This routing approach also appears to be effective, outperforming the basic pipeline. Its primary advantage lies in its **flexibility and training-free**, making it easier to integrate new pipelines in the future. However, it introduces **non-negligible extra token costs** due to LLM API calls, which weakens its cost-efficiency. Additionally, its routing performance is **less effective compared to fine-tuned smaller models**.
> >
> > > ###  Summary of Routing Approaches
> >
> > Therefore, we think routers we explored in manuscript and your suggested show two different but both practical approaches of router implementation. They are **complementary**, each with their own trade-offs, advantages and limitations.
> >
> > - **Routers in the manuscript (Tiny LLM with SFT, Cascading, DPO):**
> >     - Advantages:
> >         1. Highly effective in reducing token costs without compromising performance.
> >         2. Lightweight (0.5B or below), making them easy and affordable to deploy.
> >     - Limitations:
> >         1. Less flexible for integrating new pipelines, though cascading helps mitigate this to some extent.
> > - **Training-free LLM Routing:**
> >     - Advantages:
> >         1. Strong reasoning capabilities enable routing without specific training, making it more flexible to extend to future pipelines.
> >     - Limitations:
> >         1. Introduces significant extra LLM token costs, reducing overall efficiency.
> >         2. Less effective than fine-tuned models for routing.
> >
> > ---
> >
> > We would like to emphasize that the **core focus** of our work is **not to develop a new state-of-the-art routing algorithm**, but to **explore how routing can improve the cost-effectiveness for Text-to-SQL**.
> >
> > However, beyond this, we are truly grateful for your insightful comments have helped us further investigate this dimension, leading to additional experiments and a deeper comparative analysis, which are invaluable in refining our paper.
> >
> > We sincerely hope that these clarifications and revisions address your concerns and encourage you to reconsider the evaluation. Please don’t hesitate to reach out if you have any further questions or feedback—we would be happy to discuss them further.

---

> > > ### Comment · Reviewer_367x · 2025-06-06
> > >
> > > >On Generalization and Extensibility
> > >
> > > >Meanwhile, we fully understand and appreciate your concerns on generalization capability to new pipelines in future. In our paper, that is the reason why we introduce cascading routers.
> > >
> > > But cascading is not performing competitively in the above results, e.g. RoBERTa Casc. has 74.21 on Spider-Realistic dev.
> > > By the way, to really understand the generalization capability, we can test some datasets that have realistic settings, for example, BEAVER [1].
> > >
> > > [1] BEAVER: An Enterprise Benchmark for Text-to-SQL
> > >
> > > >We would like to emphasize that the core focus of our work is not to develop a new state-of-the-art routing algorithm, but to explore how routing can improve the cost-effectiveness for Text-to-SQL.
> > >
> > > As showed in the above reviews, some related works are not well considered or discussed in the main paper, if we want to focus on "how routing can improve the cost-effectiveness for Text-to-SQL", we need in-depth comparison with these works and highlight the main contributions.
> > >
> > > I would prefer resubmission and re-evaluation of this work after including all the discussion points.

---

> > ### Author Response · Authors · 2025-06-07
> >
> > > #### But cascading is not performing competitively in the above results ...
> >
> > We would like to clarify that both the Qwen and RoBERTa cascading routers demonstrate **very competitive performance** on the Bird-dev and Spider-dev datasets. In particular, the RoBERTa cascading router *ranks second* only to the best-performing router, while also *offering the advantage of greater extensibility*. On the Spider-realistic dataset, the Qwen cascading router also achieves a notable reduction in token usage without compromising task performance, ranking at 2nd.
> >
> > As shown in our experimental results, the training-free LLM router indeed shows some effectiveness. However, in contrast, it does not match the performance of fine-tuned routers and introduces additional token costs.
> >
> > These findings highlight the **trade-offs** between generalization/extensibility and routing performance, and **no single router perfectly fits across all scenarios.**
> >
> > >  #### BEAVER dataset
> >
> > We appreciate your suggestion on the new OOD dataset.
> >
> > In our study, we have reported OOD generalization results on Spider and Spider-realistic, both of which are widely used and well-validated benchmarks in the text-to-SQL community.
> >
> > While BEAVER is an interesting new dataset, **it has not yet been widely adopted for evaluation in the field (its leaderboard is even empty)**. Additionally, it requires a different test environment (MySQL) and is **less well-documented** compared to Spider and Bird. Due to these constraints, also with funding and time constraints, we are currently unable to test another dataset for OOD purposes, *but we actually had cited this interesting paper in the original manuscript.*
> >
> > We believe existing OOD experiments during the discussion period can present the OOD generalization of our framework, please refer to them.
> >
> >
> > Thanks again for your follow-up feedback! If you have follow-up questions, please feel free to continue the discussion with us, and we would be glad to discuss them further.

---

> > > ### Comment · Reviewer_367x · 2025-06-07
> > >
> > > >While BEAVER is an interesting new dataset, it has not yet been widely adopted for evaluation in the field (its leaderboard is even empty).
> > >
> > > I don't think this is a good reason to avoid BEAVER, some popular datasets (e.g. Spider, BIRD) may have the train/dev sets exposed to LLM, the corresponding Text2SQL performance can be inflated. By the way, Spider-test set is also available and unlikely seen by LLM, it will be a better choice than Spider-dev to demonstrate OOD performance.

---

> > ### Author Response · Authors · 2025-06-09
> >
> > We thank you for your continued engagement and suggestions.
> >
> > ---
> >
> > We would like to clarify that we do not intend to avoid the BEAVER dataset. We believe that BEAVER is a great work and can be useful in some scenarios for text-to-SQL. However, we hope to kindly note that **it has not yet been officially published or endorsed through peer-reviews.** While we believe in its potential, we think it is *not yet ready* to be used in a standard evaluation, and it would be premature to experiment with it at this stage.
> >
> > Regarding the concern of potential data leakage, we agree that this is a relevant and common challenge in LLM-based research. Investigations into the presence and impact of such leakage—as well as potential mitigation strategies—are indeed needed. However, we believe that such an investigation falls outside the scope of this particular paper.
> >
> > ---
> >
> > That said, we really appreciate your concerns and we have taken additional steps to further address them. Specifically, **following your suggestions, we conducted supplementary experiments on the Spider test set** to serve as another form of out-of-distribution evaluation.
> >
> > Our method remains robust in this evaluation setting, with particularly competitive results from the Qwen DPO router and the two cascading routers, which are even more effective than their performance on Spider dev.
> >
> > | Category    | Method             | Spider-test EX | $\bar{T}$ | TEP (×10⁻²) |
> > | ----------- | ------------------ | -------------- | --------- | ----------- |
> > | **Base**    | Basic              | 74.29          | 376.14    | -       |
> > |             | Intermediate       | *76.90*          | 2422.84   | 0.645       |
> > |             | Advanced           | **77.69**          | 3584.01   | 0.537       |
> > | **Routing** | KNN                | 75.08          | 974.26    | 0.669       |
> > |             | RoBERTa SFT        | 75.78          | 1130.04   | 1.001       |
> > |             | Qwen SFT           | 75.55          | 2196.56   | 0.350       |
> > |             | RoBERTa Cascading  | 76.90          | 1539.21   | *1.136*     |
> > |             | Qwen Cascading     | *77.04*        | 3296.22   | 0.477       |
> > |             | Qwen Pairwise Rank | 75.08          | 2245.64   | 0.214       |
> > |             | Qwen DPO           | **77.97**      | 1850.88   | **1.263**   |
> >
> > ---
> >
> > We hope these additional results help address your concerns. If you have any further questions, please feel free to continue the discussion with us!

---

> > > ### Author Response · Authors · 2025-06-11
> > > **Final Summary of Our Discussion**
> > >
> > > We once again thank you again for your review and continued engagement, which has been instrumental in strengthening our work.
> > >
> > > As the discussion period concludes, we wish to provide a **final summary** of our exchanges and the actions we have taken. **We believe we have thoroughly and respectfully addressed every concern raised, substantiating our claims with multiple rounds of new experiments conducted in response to your suggestions.**
> > >
> > > Following your helpful advice, we provided extensive new empirical results to address each concern:
> > >
> > > - **On OOD Generalization:** We evaluated on **Spider-dev and Spider-Realistic** to demonstrate strong generalization.
> > > - **On Benchmark Integrity:** We ran an extra evaluation on the **Spider test** set to address concerns about data leakage.
> > > - **On Latency:** We measured the time cost on Bird-dev, revealing that our best routers also reduce latency by over 40%, while cascading routers offer extensibility at the cost of increased processing time.
> > > - **On Extensibility:** Following your suggestions, we implemented and tested a **training-free LLM router** to analyze its routing performance and flexibility, clarifying its practical trade-offs.
> > >
> > > The results from all these new evaluations consistently validate our central claim: complexity-aware routing can practically achieve significant improvements in overall efficiency (in both tokens and time cost) without sacrificing performance.
> > >
> > > >  *We believe this extensive new evidence thoroughly addresses the issues raised, and we sincerely appreciate your time and effort in reviewing our paper, as well as your beneficial insights and suggestions once again.*

---

> ### Author Response · Authors · 2025-06-02
> **Stage I Response (1/2)**
>
> Thank you very much for your time and constructive feedback. Below, we will try our best to address the questions raised, hoping to clarify your concerns.
>
> > ### Question #1 on Out-of-domain Dataset
>
> Thank you for suggesting additional experiments on other datasets. We agree that evaluating performance on out-of-domain (OOD) datasets is valuable. Therefore, we conducted experiments on **Spider-Dev** and **Spider-Realistic Dev**, two widely adopted text-to-SQL datasets.
>
> - Please note: To evaluate OOD performance, the routers were trained **only** on the BIRD training set and then directly deployed on Spider-dev **without** any fine-tuning on Spider-train.
>
> The experiments on Spider-dev and Spider-Realistic dev continue to demonstrate the effectiveness of our routing methods, and the evaluation on this out-of-distribution dataset also shows their generalization capabilities. Most of our routers exhibit improved cost-efficiency. Notably, on Spider-dev, the Qwen DPO and RoBERTa cascading routers achieve performance comparable to consistently using the most advanced methods while reducing token consumption by over 50% and 60%, respectively; on Spider-Realistic dev, Qwen SFT, Qwen Cascading, and Qwen DPO routers reduce token consumption by 46%, 8%, and 55%, respectively, demonstrating impressive cost-efficiency and strong generalizability.
>
> **1. Experiments on Spider-dev**
>
> | Category | Method             | Spider-dev EX | $\bar{T}$ (avg token cost) | TEP(x10^-2) |
> | :------- | :----------------- | :------------ | :------------------------- | :---------- |
> | Base     | Basic              | 74.27         | 363.40                     | -           |
> |          | Intermediate       | *75.73*       | 2432.13                    | 0.345       |
> |          | Advanced           | **76.60**     | 3826.14                    | 0.356       |
> | Routing  | KNN                | 75.05         | 1033.28                    | 0.570       |
> |          | RoBERTa SFT        | 75.92         | 1222.21                    | **0.940**   |
> |          | Qwen SFT           | 75.63         | 2014.60                    | 0.403       |
> |          | RoBERTa Casc.      | *76.21*       | 1516.21                    | *0.823*     |
> |          | Qwen Casc.         | 75.53         | 3410.96                    | 0.202       |
> |          | Qwen Pairwise Rank | 74.76         | 2334.71                    | 0.122       |
> |          | Qwen DPO           | **76.31**     | 1906.79                    | 0.647       |
>
> **2. Experiments on Spider-Realistic dev**
>
> | Category | Method             | Spider-Realistic-dev EX | $\bar{T}$ (avg token cost) | TEP(x10^-2) |
> | :------- | :----------------- | :---------------------- | :------------------------- | :---------- |
> | Base     | Basic              | 73.23                   | 377.95                     | -           |
> |          | Intermediate       | *74.80*                 | 2520.67                    | 0.378       |
> |          | Advanced           | **75.39**               | 3845.75                    | 0.321       |
> | Routing  | KNN                | 72.64                   | 1171.93                    | -0.383      |
> |          | RoBERTa SFT        | 73.43                   | 1128.54                    | 0.138       |
> |          | Qwen SFT           | 75.39                   | 2064.06                    | *0.661*     |
> |          | RoBERTa Casc.      | 74.21                   | 1593.84                    | 0.416       |
> |          | Qwen Casc.         | **75.98**               | 3526.75                    | 0.451       |
> |          | Qwen Pairwise Rank | 74.61                   | 2379.99                    | 0.356       |
> |          | Qwen DPO           | *75.79*                 | 1721.43                    | **0.983**   |

---

> ### Author Response · Authors · 2025-06-05
>
> Dear Reviewer,
>
> Thank you so much for your time and effort in reviewing our paper. Your initial comments are very helpful in refining our paper. We have addressed your comments by (1) supplementary experiments and analysis on two out-of-distribution datasets; (2) detailed discussion with mechanisms in DIN-SQL; (3) report the latency on Bird and obtain new insights.
>
> We sincerely hope that these revisions have addressed your feedback. If you have any further questions, please do not hesitate to discuss them with us.
>
> Best regards,
>
> Authors

---

### Official Review · Reviewer_npNW · 2025-05-12

**Rating:** 6
**Confidence:** 4
**Ethics Flag:** 1

**Summary:**

This submission introduces a complexity-aware routing framework that assigns natural language queries to suitable text2sql generation models according to their complexity, with an aim to balance the system performance and inference costs. They also introduce a token elasticity of performance metric to measure the efficiency of generation, which measures how responsive one variable is to changes in another. The submission conducts experiments on the BIRD dataset with Roberta and Qwen2.5-0.5B as the baselines and foundation models for training routers, and gpt--4o-mini as the foundation model for SQL generation. Experiments show that the routing method can achieve a competing method compared with the direct SQL generation, but are more efficient in terms of generated tokens.

**Reasons To Accept:**

1. The method is well-motivated and the paper is generally well-written.
2. The performance on cost-efficiency is greatly improved.

**Reasons To Reject:**

1. It seems that the method can be applied to a larger scope of the application, including reasoning tasks. While I understand the choice of the task may be of the authors' interests, I still think the scope is somehow limited under the theme of COLM if the authors are focusing on the specific text2sql task, instead of the general cost-efficiency of LLMs on more tasks.
2. Having said that, the experiments only focus on one dataset, and mainly one LLM (qwen2.5-0.5B) for the experiments. More datasets should be included. Also, since the cost-efficiency mainly focuses on the tokens that are generated instead of model sizes, larger LLMs should be included for comparison.

---

> ### Author Response · Authors · 2025-06-02
> **Stage I Response (2/2)**
>
> > ### Question #3 on Various Router Implementations
>
> We actually include various router implementations in our experiments: beyond Qwen2.5-0.5B (a representative small casual LLM) with supervised fine-tuning, cascading, and direct preference optimization, we also include investigations on RoBERTa-base (a representative encoder-only model, showcasing a different model architecture) as well. Moreover, we also use KNN as a straightforward heuristic approach for comparison.
>
> > ### Question #4 on Larger LLMs as Backbone
>
> To balance experimental expenses, we deploy the more affordable GPT-4o-mini as the backbone model for a fair evaluation. Following your suggestions, to investigate a larger LLM as the backbone for a comparison, we test with a larger *GPT-4o*. *Considering the significant cost of using GPT-4o as the backbone in our task, we test with three of our representative routers due to time and funding constraints.*
>
> As shown in the table below, the tested routers are also proven effective when we deploy a larger LLM as a backbone model (GPT-4o here). RoBERTa Cascading router and Qwen DPO router reduce token consumption by 41% and 44%, respectively, without sacrificing performance, significantly improving the TEP metric. This also suggests good model transferability of our framework.
>
> As demonstrated in the table below, the tested routers also proved effective when using a larger LLM (GPT-4o here) as the backbone. Specifically, the RoBERTa Cascading router and the Qwen DPO router reduced token consumption by 41% and 44%, respectively, without sacrificing any performance, thereby significantly improving the TEP metric. This result also suggests good model transferability for our framework.
>
> | Category | Method        | Bird-dev EX | $\bar{T}$ (avg token cost)| TEP(x10^-2) |
> | :------- | :------------ | :---------- | :-------- | :---------- |
> | Base     | Basic         | 56.13       | 768.03    | -           |
> |          | Intermediate  | *57.82*     | 9357.91   | 0.269       |
> |          | Advanced      | **58.34**   | 10360.15  | 0.315       |
> | Routing  | RoBERTa SFT   | 58.15       | 4426.60   | *0.755*     |
> |          | RoBERTa Casc. | *58.87*     | 6073.55   | 0.706       |
> |          | Qwen DPO      | **59.39**   | 5832.57   | **0.881**   |
>
> Thanks again for your insightful feedback. We hope our response helps clarify and address your concerns. If you have follow-up questions, please feel free to continue the discussion with us.

---

> ### Author Response · Authors · 2025-06-02
> **Stage I Response (1/2)**
>
> Thank you very much for your time and constructive feedback. Below, we will try our best to address the questions raised, hoping to clarify your concerns.
>
> > ### Question #1 on Broader Applications
>
> We appreciate you highlighting the potential for our approach to be applied to broader reasoning tasks. It is true that our method could potentially be adopted for other general reasoning tasks, such as math reasoning, to reduce computational costs.
>
> However, not all reasoning tasks are as token-consuming as text-to-SQL. Text-to-SQL is special because of its close relationship with databases: a large database can have a very long and complicated schema, making it extremely token-consuming and challenging for LLMs to analyze and generate correct SQL queries. It can cost *tens of thousands of tokens or even more for LLM to just deal with a single query* in the Bird dataset, and often, multiple times of token investment may yield only marginal improvements (~1%) on leaderboards.
>
> Also considering the critical role of text-to-SQL in democratizing database access for novice users, we think there is an *urgent need* for more sustainable text-to-SQL approaches to enhance its practicality, which motivates us to explore and focus on this special and important field.
>
> > ### Question #2 on Experiments for More Datasets
>
> Thank you for suggesting additional experiments on other datasets. We agree that evaluating performance on out-of-domain (OOD) datasets is valuable. Therefore, we conducted experiments on **Spider-Dev** and **Spider-Realistic Dev**, two widely adopted text-to-SQL datasets.
>
> - Please note: To evaluate OOD performance, the routers were trained **only** on the BIRD training set and then directly deployed on Spider-dev **without** any fine-tuning on Spider-train.
>
> The experiments on Spider-dev and Spider-Realistic dev continue to demonstrate the effectiveness of our routing methods, and the evaluation on this out-of-distribution dataset also shows their generalization capabilities. Most of our routers exhibit improved cost-efficiency. Notably, on Spider-dev, the Qwen DPO and RoBERTa cascading routers achieve performance comparable to consistently using the most advanced methods while reducing token consumption by over 50% and 60%, respectively; on Spider-Realistic dev, Qwen SFT, Qwen Cascading, and Qwen DPO routers reduce token consumption by 46%, 8%, and 55%, respectively, demonstrating impressive cost-efficiency and strong generalizability.
>
> **1. Experiments on Spider-dev**
>
> | Category | Method             | Spider-dev EX | $\bar{T}$ (avg token cost) | TEP(x10^-2) |
> | :------- | :----------------- | :------------ | :------------------------- | :---------- |
> | Base     | Basic              | 74.27         | 363.40                     | -           |
> |          | Intermediate       | *75.73*       | 2432.13                    | 0.345       |
> |          | Advanced           | **76.60**     | 3826.14                    | 0.356       |
> | Routing  | KNN                | 75.05         | 1033.28                    | 0.570       |
> |          | RoBERTa SFT        | 75.92         | 1222.21                    | **0.940**   |
> |          | Qwen SFT           | 75.63         | 2014.60                    | 0.403       |
> |          | RoBERTa Casc.      | *76.21*       | 1516.21                    | *0.823*     |
> |          | Qwen Casc.         | 75.53         | 3410.96                    | 0.202       |
> |          | Qwen Pairwise Rank | 74.76         | 2334.71                    | 0.122       |
> |          | Qwen DPO           | **76.31**     | 1906.79                    | 0.647       |
>
> **2. Experiments on Spider-Realistic dev**
>
> | Category | Method             | Spider-Realistic-dev EX | $\bar{T}$ (avg token cost) | TEP(x10^-2) |
> | :------- | :----------------- | :---------------------- | :------------------------- | :---------- |
> | Base     | Basic              | 73.23                   | 377.95                     | -           |
> |          | Intermediate       | *74.80*                 | 2520.67                    | 0.378       |
> |          | Advanced           | **75.39**               | 3845.75                    | 0.321       |
> | Routing  | KNN                | 72.64                   | 1171.93                    | -0.383      |
> |          | RoBERTa SFT        | 73.43                   | 1128.54                    | 0.138       |
> |          | Qwen SFT           | 75.39                   | 2064.06                    | *0.661*     |
> |          | RoBERTa Casc.      | 74.21                   | 1593.84                    | 0.416       |
> |          | Qwen Casc.         | **75.98**               | 3526.75                    | 0.451       |
> |          | Qwen Pairwise Rank | 74.61                   | 2379.99                    | 0.356       |
> |          | Qwen DPO           | *75.79*                 | 1721.43                    | **0.983**   |

---

> ### Comment · Reviewer_npNW · 2025-06-03
>
> Thanks to the authors for their clarification.
> I am happy to raise my scores.

---

> > ### Author Response · Authors · 2025-06-03
> >
> > Thank you for your valuable feedback! Should you have any further questions during the second stage of discussions, we would be more than happy to address them.

---

### Official Review · Reviewer_8cMx · 2025-05-12

**Rating:** 8
**Confidence:** 4
**Ethics Flag:** 1

**Summary:**

This paper introduces EllieSQL, a novel complexity-aware routing framework for Text-to-SQL tasks that explicitly addresses the under-discussed challenge of computational inefficiency in current LLM-based systems. Instead of treating all queries equally, EllieSQL estimates the complexity of each natural language query and dynamically assigns it to one of three tiered SQL generation pipelines: Basic, Intermediate, or Advanced. These pipelines trade off between performance and token usage.

**Reasons To Accept:**

1. The paper addresses an often-neglected yet pressing challenge in Text-to-SQL systems—computational cost and inefficiency—which is a major barrier for real-world deployment. The focus on cost-effectiveness adds significant practical value to the field.
2. EllieSQL introduces a complexity-aware routing mechanism that dynamically directs queries to different SQL generation pipelines based on estimated difficulty. This pipeline-level routing is novel and distinct from existing model-level routing approaches.
3. The proposed Token Elasticity of Performance provides an insightful and principled way to measure the cost-efficiency tradeoff, complementing traditional accuracy metrics and encouraging more sustainable model use.
4. Experiments on the Bird benchmark show that EllieSQL—especially with the Qwen DPO router—achieves significant token reduction without sacrificing performance, confirming both its effectiveness and efficiency.

**Reasons To Reject:**

The router in EllieSQL is tightly coupled with specific downstream SQL generation pipelines, making the approach difficult to generalize or directly apply to other Text-to-SQL systems. Adapting the framework to new tasks or models requires retraining the router, which introduces non-trivial annotation and engineering costs. While a similar automatic labeling strategy could be reused, many real-world Text-to-SQL scenarios may lack sufficient training data to support this. This limits the method’s modularity and practical applicability. A more generalizable or end-to-end trainable routing design would significantly improve usability.

---

> ### Author Response · Authors · 2025-06-02
>
> Thank you very much for your time and constructive feedback. As we mentioned in the limitation section,  labeling training data indeed introduces an upfront cost (although this expense becomes increasingly amortized with wide adoption).
>
> For scenarios with limited training data, we think our cascading routers offer viable options. This is because for cascading structured routers, integrating a new method only necessitates the addition of a corresponding binary router, without requiring retraining of the original router. Consequently, this approach facilitates easier extension and integration while simultaneously reducing the demand for extensive training data. Therefore, we agree that pursuing more scalable and generalizable routing for text-to-SQL is an important topic for future studies.
>
> We appreciate your insightful feedback once again.

---

> > ### Comment · Reviewer_8cMx · 2025-06-05
> >
> > Thanks to the authors for their response.

---

### Official Review · Reviewer_b7ZP · 2025-05-25

**Rating:** 6
**Confidence:** 4
**Ethics Flag:** 1

**Summary:**

The paper targets the problem of high computational costs in systems using advanced LLM-based text-to-sql approaches, which hinder their practicality for real-world deployment. It proposes EllieSQL, a complexity-aware routing framework that assigns queries to SQL generation pipelines based on their estimated difficulty, using lightweight pipelines for simple queries and computationally intensive pipelines for complex ones, along with a new efficiency metric called Token Elasticity of Performance (TEP). Experiments show that EllieSQL reduces token usage by over 40% without compromising accuracy and more than doubles cost-effciency (TEP) compared to always using the most complex pipelines.

**Reasons To Accept:**

This paper targets an important problem, while prior work has explored routing different LLMs for prompts, routing within Text-to-SQL pipelines remains under-explored.

The idea of routing based on the results of schema linking makes a lot of sense, as it provides a natural signal of query complexity that can guide the selection of appropriate generation pipelines.

The results seem promising, showing significant reductions in token usage without compromising accuracy.

**Reasons To Reject:**

The routing decision relies heavily on schema linking outcomes, yet schema linking itself is a challenging task. This dependency introduces potential fragility, if schema linking fails or produces noisy results, the entire routing strategy may misclassify query complexity.

The paper does not compare against existing LLM routing methods such as RouteLLM, which could also offer competitive performance in this setting. For example, the baseline can simply treat queries as standard prompts and use RouteLLM, or treat queries plus table schema or samples (potentially filter with retrieval) as the input to RouteLLM. It thereby opens the question of how much EllieSQL actually improves over simpler or existing alternatives.

---

> ### Author Response · Authors · 2025-06-02
>
> Thank you very much for your time and constructive feedback. Below, we will try our best to address the questions raised, hoping to clarify your concerns.
>
> > ### Question #1 on Schema Linking
>
> 1. Perfect schema linking is indeed challenging, its primary goal is to identify relevant tables and columns for the current task. As long as we maintain a high recall, some noise is tolerable. This is because missing a column will negatively impact SQL generation, whereas a redundant column can often be ignored during the SQL generation stage. This makes redundant noise more acceptable. In this regard, recall is prioritized. We examined the schema linking performance of GPT-4o-mini in our experiments and found that it achieves a high recall of 93% while reducing the number of columns by over 90% on average. This suggests that the schema linking in our experiments is sufficiently robust.
> 2. Moreover, as schema linking is a standard procedure in text-to-SQL, we are motivated to utilize the results, which is why we position the routers after schema linking rather than before it. The database schema in the Bird dataset can be quite large, making it very difficult to route tasks solely based on the query and the full database schema. In contrast, since schema linking is a standard procedure, its intermediate results are valuable. Although these results may contain some noise from redundant elements, we still filter out over 90% of columns on average. This provides much more informative references for the routers compared to the full database schema. Therefore, we believe that placing routers right after schema linking is an appropriate design choice to utilize the linked schema.
>
> > ### Question #2 on Routing
>
> Thank you for highlighting the relevant work of RouteLLM; we have cited it in the related work section. RouteLLM is an illuminating work on LLM routing and inspired some of our router implementations.
>
> RouteLLM proposes supervised fine-tuning (SFT) open-source LLMs to perform binary selection of the more suitable LLM using a special classification token. Our classification-based routers trained through SFT are similar to this approach. However, our method differs in that it supports multi-class classification by adding a linear classifier head. Furthermore, we introduced Direct Preference Optimization (DPO) to further help routers learn preferences, and these DPO-trained routers demonstrate better routing performance compared with SFT-trained ones. They are also the most effective routers in our studies. We believe that investigating and integrating more advanced routers could be an interesting topic for future research.
>
>
> Thanks again for your insightful feedback. We hope our response helps clarify and address your concerns. If you have follow-up questions, please feel free to continue the discussion with us.

---

> > ### Comment · Reviewer_b7ZP · 2025-06-09
> > **Thanks for your response**
> >
> > Thanks for your response. The schema linking results are helpful. I will increase the score a bit.

---

> > > ### Author Response · Authors · 2025-06-09
> > >
> > > We sincerely appreciate your positive feedback! Should you have any further questions during the remaining discussion time, we would be more than happy to address them.

---

> ### Author Response · Authors · 2025-06-06
>
> Dear Reviewer,
>
> Thank you so much for your time and effort in reviewing our paper. Your initial comments are very helpful in refining our paper. We evaluate the schema linking performance in our experiments, which is robust. We also clarify the position of routers, as well as the connection and the difference between RouteLLM and our SFT-based routers.
>
> **Additionally, we conducted further experiments during the discussion period, which are summarized in the general response. We kindly invite you to refer to those results if you are interested in.**
>
> We sincerely hope that these revisions have addressed your feedback. If you have any further questions, please do not hesitate to discuss them with us.
>
> Best regards,
>
> Authors

---

### Author Response · Authors · 2025-06-06
**General Response**

We greatly appreciate the efforts of all reviewers in reviewing our paper and providing insightful suggestions for its improvement.

We are delighted and encouraged that our topic is found to be important, well-motivated, novel, and practical; and our experimental results are thought promising. Reviewers raised helpful suggestions, and below we report additional experiments to address these concerns. **We have also uploaded separate responses for each reviewer; please refer to them for more specific clarifications and analysis.**

### 1. Out-of-Domain Dataset

Following reviewers npNW and 367x's suggestions, we evaluated our routers on Spider-Dev and Spider-Realistic Dev sets, without fine-tuning on their training set. The results confirm generalization and cost-efficiency across datasets.

| Dataset                  | Category | Method             | EX    | $\bar{T}$ | TEP (×10⁻²) |
| ------------------------ | -------- | ------------------ | ----- | --------- | ---------------------- |
| **Spider-Dev**           | Base     | Basic              | 74.27 | 363.40    | --                     |
|                          |          | Intermediate       | 75.73 | 2432.13   | 0.345                  |
|                          |          | Advanced           | 76.60 | 3826.14   | 0.356                  |
|                          | Routing  | KNN                | 75.05 | 1033.28   | 0.570                  |
|                          |          | RoBERTa SFT        | 75.92 | 1222.21   | 0.940                  |
|                          |          | Qwen SFT           | 75.63 | 2014.60   | 0.403                  |
|                          |          | RoBERTa Casc.      | 76.21 | 1516.21   | 0.823                  |
|                          |          | Qwen Casc.         | 75.53 | 3410.96   | 0.202                  |
|                          |          | Qwen Pairwise Rank | 74.76 | 2334.71   | 0.122                  |
|                          |          | Qwen DPO           | 76.31 | 1906.79   | 0.647                  |
| **Spider-Realistic Dev** | Base     | Basic              | 73.23 | 377.95    | --                     |
|                          |          | Intermediate       | 74.80 | 2520.67   | 0.378                  |
|                          |          | Advanced           | 75.39 | 3845.75   | 0.321                  |
|                          | Routing  | KNN                | 72.64 | 1171.93   | -0.383                 |
|                          |          | RoBERTa SFT        | 73.43 | 1128.54   | 0.138                  |
|                          |          | Qwen SFT           | 75.39 | 2064.06   | 0.661                  |
|                          |          | RoBERTa Casc.      | 74.21 | 1593.84   | 0.416                  |
|                          |          | Qwen Casc.         | 75.98 | 3526.75   | 0.451                  |
|                          |          | Qwen Pairwise Rank | 74.61 | 2379.99   | 0.356                  |
|                          |          | Qwen DPO           | 75.79 | 1721.43   | 0.983                  |

### 2. Larger Backbone LLM

Per reviewer npNW's suggestion, we tested GPT-4o as the backbone on Bird-dev, further demonstrating model transferability and sustained efficiency with larger models.

| Method        | EX    | $\bar{T}$ | TEP (×10⁻²) |
| ------------- | ----- | --------- | ---------------------- |
| Basic         | 56.13 | 768.03    | --                     |
| Advanced      | 58.34 | 10360.15  | 0.315                  |
| RoBERTa SFT   | 58.15 | 4426.60   | 0.755                  |
| RoBERTa Casc. | 58.87 | 6073.55   | 0.706                  |
| Qwen DPO      | 59.39 | 5832.57   | 0.881                  |

### 3. Latency Evaluation

Following reviewer 367x's suggestion, we measured time duration on Bird-dev, observing that cascading routers offer better extensibility at the cost of increased latency.

| Method        | EX    | $\bar{T}$ | TEP (×10⁻²) | Time (s) |
| ------------- | ----- | --------- | ---------------------- | -------- |
| Basic         | 51.83 | 695.55    | --                     | 114      |
| Advanced      | 55.02 | 13002.91  | 0.348                  | 653      |
| Qwen DPO      | 55.41 | 7641.51   | 0.692                  | 396      |
| RoBERTa Casc. | 55.15 | 8211.00   | 0.593                  | 467      |

### 4. Training-Free LLM-Based Routing

To address reviewer 367x's concern on extensibility, we added a training-free LLM router and analyzed trade-offs of both finetuned tiny LLM-based routers and large training-free LLM implementations.

| Method      | Bird Dev EX | $\bar{T}$ | TEP (×10⁻²) |
| ----------- | ----------- | --------- | ----------- |
| LLM Routing | 53.78       | 11551.42  | 0.241       |


> We hope these results address the reviewers’ feedback and are happy to discuss further or revise the manuscript accordingly. Once again, many thanks to all the reviewers!

---

### Author Response · Authors · 2025-06-11
**Concluding Summary: New Actions from the Discussion Period**

We would like to extend our sincerest gratitude to all reviewers for their time, expertise, and constructive feedback. The detailed discussion period has been invaluable to our manuscript.

We are encouraged that the reviewers found our work to be novel, well-motivated, promising, and of high practical importance for the field of Text-to-SQL. We are especially grateful for the insightful suggestions that prompted us to conduct a comprehensive set of additional experiments.

As the discussion period concludes, we wish to provide a **concluding summary** of our exchanges and the actions we have taken during the whole discussion period. In response to the reviewers' collective feedback, we have:

- **Evaluated Out-of-Domain Generalization** on Spider-dev and Spider-Realistic.
- **Measured Latency/Time Cost**, demonstrating that our best routers improve not only token cost but also processing speed.
- **Tested with a Larger Backbone LLM** to confirm the transferability of our framework to lager backbone models.
- **Explored Extensibility** by implementing a training-free LLM router to analyze its practical trade-offs.
- **Verified Schema Linking and Related Work** by providing performance details of schema linking and clarifying the novelties of our approach.
- **Addressed Benchmark Integrity** by running a final set of experiments on the **Spider test set**.

**We believe that our detailed responses and these extensive new results have thoroughly addressed every concern raised**, leading to positive feedback from reviewers. The thoughtful discussions and supplementary experiments have made our paper more robust and complete.

Thanks a lot once again for all reviewers' dedication to the review process!

---

### Decision · Program_Chairs · 2025-07-08

**Decision:**

Accept

**Comment:**

This paper describes an efficiency-improving method for text-to-SQL processing in which a router decides whether a question can be answered by a lightweight model or requires a more expensive LLM call. Experiments, including additional results provided in response to reviewer concerns, demonstrate the effectiveness of the approach. Analysis addresses key questions about trade-offs in different variants of the approach.

Overall, this is a novel approach with positive results. The key reviewer questions have been addressed in the discussion period and adding the new results to the paper will strengthen it significantly.

Some particularly important points raised should be addressed in the final version of the paper:

- Add the new results presented in the discussion period and discuss them.
- Expand the discussion of other routing methods in section 2 to clarify where there is no direct comparison with RouteLLM (for example).
- Discuss the reliance on schema linking in the limitations section.
- Discuss other text-to-SQL methods that have some form of routing further, to distinguish this approach from the prior work.